# A Comparison of Two Supplementary Doses of Vitamin A on Performance, Intestine and Immune Organ Development, as well as Gene Expression of Inflammatory Factors in Young Hy-Line Brown Laying Pullets

**DOI:** 10.3390/ani12101271

**Published:** 2022-05-15

**Authors:** Qinliang Chen, Xiaoqing Han, Huiling Zhu, Yulan Liu, Xiao Xu

**Affiliations:** 1Hubei Key Laboratory of Animal Nutrition and Feed Science, Wuhan Polytechnic University, Wuhan 430023, China; chenqinliang123@gmail.com (Q.C.); zhuhuiling@whpu.edu.cn (H.Z.); liuyulan@whpu.edu.cn (Y.L.); 2Wuhan Hualuo Branch, China Animal Husbandry Industry Co., Ltd., Wuhan 430050, China; pumpkin_h@163.com

**Keywords:** immune, intestine, vitamin A, young laying pullets

## Abstract

**Simple Summary:**

In young laying pullets, vitamin A (VA) is a necessary micronutrient that plays an important role in intestinal integrity, cell proliferation and differentiation, and modulation of the immune system. However, the recommended dose for young laying pullets in NRC (1994) is too low to use under practical conditions. Both the feed industry and producers often prefer to supplement diets for chicks with more than 6000 IU/kg VA to improve performance and alleviate negative stresses currently. Therefore, the objective of this study was to compare two supplementary doses (6000 vs. 12,000 IU/kg) of vitamin A on performance, development of intestine and immune organs, as well as gene expression of inflammatory factors in young Hy-Line Brown laying pullets. The results showed that supplementary 12,000 IU/kg of VA improved performance and intestine and immune organ development, and alleviated gene expression of inflammatory factors in young Hy-Line Brown laying pullets.

**Abstract:**

The objective of this study was to compare two supplementary doses (6000 vs. 12,000 IU/kg) of vitamin A (VA) on the performance, development of intestine and immune organs, as well as gene expression of inflammatory factors in young Hy-Line Brown laying pullets. A total of 288 one-day-old Hy-Line Brown laying pullets (weighing 42.15 ± 0.23 g) were allotted into two treatments with 12 replicate cages and 12 birds per cage. During the 35-day period, the pullets were fed a basal diet supplemented with different doses of VA (6000 IU/kg VA in control group; 12,000 IU/kg VA in treatment group), respectively. The results showed that supplementary high doses of VA reduced the feed-to-gain ratio from day 21 to 35 (*p* < 0.05). Moreover, the pullets fed high doses of VA diets had increased length and relative weight of duodenum, jejunum, and ileum (*p* < 0.05). From observations on morphology, high doses of VA diets increased the villus height and the ratio of villus height to crypt depth in the jejunum and ileum (*p* < 0.05). High doses of VA diets also increased the relative weight of immune organs (*p* < 0.05). Furthermore, the gene expressions of inflammatory factors were decreased in the thymus of the pullets fed high doses of VA diets (*p* < 0.05). In summary, supplementary 12,000 IU/kg doses of VA improved performance and intestine and immune organ development, and alleviated gene expressions of inflammatory factors in young Hy-Line Brown laying pullets.

## 1. Introduction

Immune stress appears to be a potential factor that causes lower performance and health status in young laying pullets. In this case, the supplementation of dietary vitamins may be effective to improve immune status and performance. In young laying pullets, vitamin A (VA) is a necessary micronutrient, and the recommended dose for young laying pullets in NRC (1994) is 1420 IU/kg. However, under practical conditions, both the feed industry and producers often prefer to supplement diets for chicks with more than 6000 IU/kg VA to improve performance and alleviate negative stresses [1].

VA plays an important role in intestinal integrity, cell proliferation and differentiation, and modulation of the immune system [2,3]. VA plays a regulatory role in cell- and antibody-mediated immune responses by modulating T lymphocyte activation and proliferation as well as B lymphocyte proliferation and antibody production [4]. Moreover, VA plays a key role in mediating innate defenses by promoting differentiation and maturation of epithelial cells and formation of the epithelial layer [5]. Modern laying hen production usually faces a variety of stressful environments; in this case, the recommendation of young laying pullets dietary VA by NRC (1994) is low, and higher doses of dietary VA could improve performance and immune status. There was a study demonstrating that supplementation with dietary VA increased growth performance and immune status in chicks [6].

Interestingly, no work has been done to evaluate the effects of dietary VA on performance and immune status in young laying pullets. Therefore, we hypothesized that the addition of VA in young laying pullets’ diets could improve the performance and immune status. Accordingly, the objective of this study was to compare the effects of two supplementary doses (6000 vs. 12,000 IU/kg) of VA on performance and immune status in young laying pullets.

## 2. Materials and Methods

### 2.1. Experimental Birds, Diets and Design

This experimental protocol (No. WPU202011001) used in this study was approved by the Institutional Animal Care and Use Committee of Wuhan Polytechnic University (Wuhan, China). A total of 288 one-day-old Hy-Line Brown laying pullets (weighing 42.15 ± 0.23 g) were purchased from the Hy-Line Poultry Breeding Company (Wuhan, China). All pullets were raised in wire-floored cages (120 × 120 × 60 cm^3^) in an environmentally controlled room with continuous light and had ad libitum access to feed and water. The ambient temperature was maintained at 36 °C at the start of experiment and was decreased as the birds progressed in age. The relative humidity was set at 45 to 55% and was kept within this range. The whole trial lasted for 35 days.

All pullets were allotted to 2 treatments and 12 replicate cages containing 12 pullets by a completely randomized design. The pullets were fed a basal diet supplemented with different doses of VA (6000 IU/kg VA in the control group; 12,000 IU/kg VA in the treatment group), respectively. The basal diets were formulated according to the nutrient requirements suggested by NRC (1994) for young laying pullets. The ingredient composition and nutrient levels of the basal diets are shown in Table 1.

### 2.2. Sample Collection

All pullets were weighed individually after their arrival from the hatchery. The pullets were also weighed on days 7, 14, 21, 28, and 35. Feed bags were weighed at the same time and these values were used to calculate average daily gain (ADG), average daily feed intake (ADFI), and feed-to-gain ratio (F/G).

Two pullets per cage were selected according to the average weight of the pullets in the cage and euthanized on days 14, 21, 28, and 35. One pullet was euthanized to collecting samples for determining intestinal morphology and gene expressions of inflammatory factors in the thymus. Immediately after euthanasia, two cm tissue samples of the small intestine were obtained from the jejunum and ileum. The intestinal samples were flushed with 0.9% salt solution, fixed with 10% formaldehyde-phosphate buffer and kept at 4 °C for microscopic assessment of intestinal morphology. The thymus and bursa of fabricins were collected and rapidly frozen in liquid nitrogen and stored at −80 °C for measurement of gene expressions. All intestinal samples were collected within 15 min of slaughter. The other pullet was euthanized to collect samples for determining the length of the small intestine (duodenum, jejunum, and ileum) and relative organ weight (duodenum, jejunum, ileum, spleen, thymus, and bursa) to live body weight according to a previous study [7].

### 2.3. Intestinal Morphology Analysis

Villus height and crypt depth were measured according to a previous study [8]. The villus height and crypt depth from 12 pullets were averaged to obtain the mean villus height and crypt depth for each treatment group.

### 2.4. Quantification of mRNA Expression of the Genes

The gene expressions of toll-like receptors 4 (TLR4), myeloid differentiation factor 88 (MyD88), nucleotide-binding oligomerization domain protein 1 (NOD1), receptor-interacting protein 2 (RIP2), and nuclear factor-κB (NF-κB) in the thymus were determined using the quantitative real-time polymerase chain reaction (qRT-PCR) analysis as described before [9]. The primer pairs for amplification of target genes are presented in Table 2. The expression of the target genes relative to housekeeping gene (β-actin) was analyzed by the 2^−^^△△CT^ method [10]. Relative mRNA abundance of each target gene was normalized to the pullets fed treatment diets.

### 2.5. Statistical Analyses

All data were analyzed using Student’s *t*-test to evaluate significant differences in values between the control and treatment groups (SAS Institute Inc., Cary, NC, USA). The cage was the experimental unit for performance and the individual pullet was the experimental unit for other parameters. Results are expressed as least squares means and standard error of the mean (SEM). The results are considered significant at *p* < 0.05.

## 3. Results and Discussion

### 3.1. Growth Performance

There was no difference in performance between the two groups in the initial 14 days (*p* > 0.05, Table 3). On days 21, 28 and 35, the body weight (BW) and average daily gain (ADG) of the young pullets in the two groups had no difference (*p* > 0.05). However, high supplementary doses of VA reduced the F/G in young Hy-Line Brown laying pullets (*p* < 0.05). The results of this study showed that high doses of VA decreased F/G in young laying pullets. This improvement indicates that nutrient utilization might be more efficient in pullets fed high doses of VA diets. Some studies reported that VA supplementation reduced F/G in young broilers [1]. Although VA supplementary doses used in this study were above those indicated by the NRC (1994), no results were found indicating that pullets had hypervitaminosis A. However, it was possible to indicate that the VA requirements of young laying pullets may be higher than the NRC (1994) recommended doses, which is in agreement with previous studies [11]. The current recommendation for VA dose in the whole industry and academia is 10,000 IU/kg for young Hy-Line Brown hens. However, the Chinese Safe Use Standard of Feed Additives (No. 2625) stipulates that the maximum dose of VA for young pullets after 14 days is 10,000 IU/kg. A majority of commercial pullet feed contains about 6000 IU/kg VA. Therefore, the current study was designed with 6000 and 12,000 IU/kg supplementary doses.

### 3.2. Intestine Length and Relative Organ Weight

On day 14, the young pullets fed high doses of VA diets had longer ileum length (*p* < 0.05, Table 4). On day 21, high levels of VA diets increased the jejunum length and relative organ weight of the duodenum (*p* < 0.05). On day 28, the young pullets fed high doses of VA diets had longer duodenum and jejunum length, as well as relative organ weight of jejunum (*p* < 0.05). On day 35, the young pullets fed high doses of VA had increased length and relative weight of the duodenum, jejunum, and ileum (*p* < 0.05). The length and relative organ weight of the intestine were increased in the pullets fed high doses of VA. This result indicates that supplementary 12,000 IU/kg doses of VA might increase the development of the gastrointestinal tract, which was evident with increasing digestive and absorption capacities. Some previous studies showed that VA could improve intestinal integrity, cell proliferation, and differentiation [12]. Moreover, Fan et al. (2015) demonstrated that VA supplementation could promote mucosal development [13]. These intestinal length and weight increases were consistent with the improving F/G in the pullets fed high doses of VA.

### 3.3. Intestinal Morphology

On days 14 and 21, the young pullets fed high doses of VA had increased jejunal villus height to crypt depth ratio (VCR) and ileal villus height and VCR (*p* < 0.05, Table 5). On day 28, the young pullets fed high doses of VA diets had increased ileal villus height and VCR (*p* < 0.05). On day 35, the young pullets fed high doses of VA diets had increased villus height and VCR both in the jejunum and ileum (*p* < 0.05). The intestinal healthy status is often reflected by structural and functional integrity. Villus height, crypt depth, and VCR have been used to quantify the intestinal morphology [14]. In the present study, high doses of VA supplementation increased intestinal VCR, which was similar to the previous findings [15]. These results indicate that supplementary 12,000 IU/kg doses of VA have a significant effect on protecting intestinal mucosa and improving intestinal repair.

### 3.4. Relative Weight of Immune Organs

On days 14 and 21, the young pullets fed high doses of VA had increased relative weight of the spleen and thymus (*p* < 0.05, Table 6). However, the relative weight of the bursa of fabricins in the young pullets fed high doses of VA were increased on days 28 and 35 (*p* < 0.05). The relative immune organ weight is an indicator of the immune status of poultry. In the present study, high doses of VA diets increased the relative weight of immune organs. The results indicate that dietary supplementation with VA improves development of immune organs. A previous study also showed that dietary supplementation with VA increased the relative bursa of fabricius weight in broilers [6,16]. VA is a necessary nutrient for improving immune response in poultry [16].

### 3.5. Gene Expressions of Inflammatory Factors in Thymus

The young pullets fed high doses of VA diets had reduced gene expressions of MyD88, NOD1, RIP2, and NF-κB in the thymus on day 14 (*p* < 0.05, Figure 1). Moreover, on days 21, 28, and 35, high doses of VA diets decreased gene expressions of TLR4, MyD88, NOD1, RIP2, and NF-κB in the thymus (*p* < 0.05). TLR4 and NOD are important protein families of inflammatory signaling pathways [17]. Activation of these pathways resulted in multiple organ damage [18]. The present study showed that high doses of VA diets alleviated key gene expression of TLR4 (TLR4 and MyD88) and NOD (NOD2 and NF-κB) pathways in the young pullets. It demonstrated that supplementation with VA had the function of alleviating inflammation. Some studies have confirmed that VA maintained animals’ health by regulating cellular immunity, humoral immunity, and nonspecific immune responses [19]. The reducing inflammatory gene expressions were in agreement with the results of increased relative weight of immune organs, which demonstrate that high dietary doses of VA improve immune function in young laying pullets.

## 4. Conclusions

In conclusion, compared with 6000 IU/kg, supplementary 12,000 IU/kg doses of VA improved performance and intestine and immune organ development, and alleviated gene expression of inflammatory factors in young Hy-Line Brown laying pullets.

## Figures and Tables

**Figure 1 animals-12-01271-f001:**
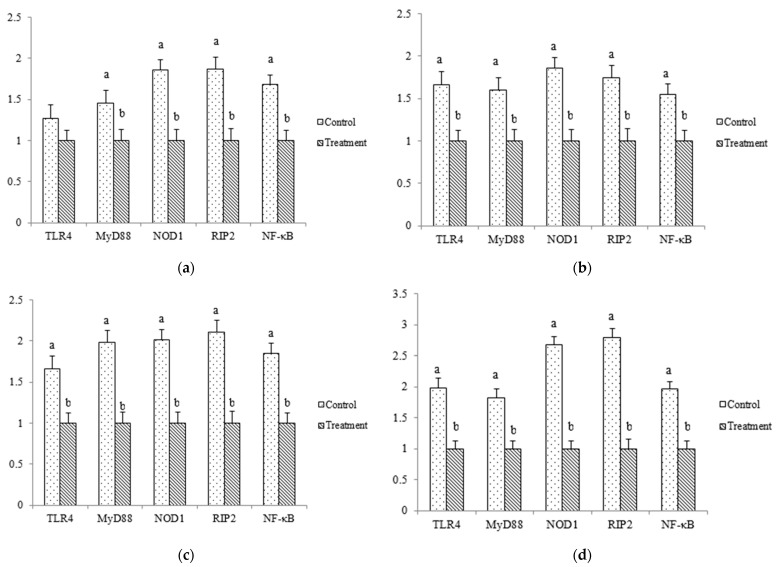
Gene expressions of TLR4, MyD88, NOD1, RIP2, and NF-κB in thymus of young laying pullets on days 14 (**a**), 21 (**b**), 28 (**c**) and 35 (**d**). Different letters indicate significant differences between mean values for a gene expression (*p* < 0.05). TLR4, toll-like receptors 4; MyD88, myeloid differentiation factor 88; NOD1, nucleotide-binding oligomerization domain protein; RIP2, receptor-interacting protein 2; NF-κB, nuclear factor-κB.

**Table 1 animals-12-01271-t001:** Composition and nutrient content of basal diets for the young laying pullets (as fed basis).

Ingredients, %		Nutrient Content, % ^2^	
Corn	56.59	Metabolizable energy, MJ/kg	12.18
Flour	3.40	Crude protein	21.45
Soybean meal	25.50	Ca	1.22
Extruded soybean	10.00	Nonphytate phosphorus	0.45
DL-methionine	0.12	Lysine	1.15
Limestone	1.41	Methionine	0.43
Calcium hydrophosphate	1.68	Methionine + Cysteine	0.82
Salt	0.30		
Vitamin and mineral premix ^1^	1.00		
Total	100.00		

^1^ Premix supplied per kg diet: vitamin D, 3025 IU; vitamin E, 30 IU; vitamin K_3_, 2.2 mg; thiamine, 1.65 mg; riboflavin, 6.6 mg; pyridoxine, 3.3 mg; cobalamin, 17.6 μg; nicotinic acid, 22 mg; pantothenic acid, 13.2 mg; folic acid, 0.33 mg; biotin, 88 μg; choline chloride, 500 mg; iron, 48 mg; zinc, 96.6 mg; manganese, 101.76 mg; copper, 10 mg; selenium, 0.05 mg; iodine, 0.96 mg; cobalt, 0.3 mg. ^2^ The nutrient content were analyzed values except metabolizable energy and nonphytate phosphorus, which were calculated values.

**Table 2 animals-12-01271-t002:** Primer sequences used for real-time PCR.

Gene	Forward (5′-3′)	Reverse (5′-3′)	Product Length (bp)	Accession Numbers
*TLR4*	GTCTCTCCTTCCTTACCTGCTGTTC	AGGAGGAGAAAGACAGGGTAGGTG	187	AY064697
*MyD88*	AGAAGGTGTCGGAGGATGGTG	GGGCTCCAAATGCTGACTGC	365	NM_001030962
*NOD1*	AGCACTGTCCATCCTCTGTCC	TGAGGGTTGGTAAAGGTCTGCT	62	JX465487
*RIP2*	CAGTGTCCAGTAAATCGCAGTTG	CAGGCTTCCGTCATCTGGTT	206	XM_003355027.1
*NF-κB*	GTGTGAAGAAACGGGAACTG	GGCACGGTTGTCATAGATGG	138	NM_205129
*β-actin*	GTGTGAAGAAACGGGAACTG	GGCACGGTTGTCATAGATGG	205	L08165

*TLR4*, toll-like receptors 4; *MyD88*, myeloid differentiation factor 88; *NOD1*, nucleotide-binding oligomerization domain protein; *RIP2*, receptor-interacting protein 2; *NF-κB*, nuclear factor-κB.

**Table 3 animals-12-01271-t003:** Effects of two supplementary doses of vitamin A on performance of young laying pullets ^1^.

Item	Control	Treatment	SEM	*p*-Value
D 0				
BW, g	42.16	42.15	0.23	0.91
D 7				
BW, g	64.96	65.29	0.67	0.92
ADG, g/d	3.80	3.86	0.12	0.90
ADFI, g/d	8.48	8.40	0.24	0.31
F/G	2.23	2.18	0.07	0.39
D 14				
BW, g	120.18	120.32	1.17	0.75
ADG, g/d	6.00	6.01	0.18	0.82
ADFI, g/d	11.08	10.61	0.32	0.18
F/G	1.85	1.77	0.05	0.23
D 21				
BW, g	206.37	206.53	2.10	0.87
ADG, g/d	8.21	8.22	0.20	0.91
ADFI, g/d	14.53	13.90	0.35	0.12
F/G	1.77 ^a^	1.69 ^b^	0.04	0.02
D 28				
BW, g	305.60	306.72	3.46	0.76
ADG, g/d	9.76	9.80	0.22	0.89
ADFI, g/d	19.05 ^a^	18.24 ^b^	0.38	0.04
F/G	1.95 ^a^	1.86 ^b^	0.04	0.03
D 35				
BW, g	408.26	414.61	5.05	0.24
ADG, g/d	10.77	10.95	0.23	0.55
ADFI, g/d	24.07 ^a^	22.73 ^b^	0.53	<0.01
F/G	2.23 ^a^	2.08 ^b^	0.05	<0.01

ADG, average daily gain; ADFI, average daily feed intake; BW, body weight; F/G, feed-to-gain ratio; SEM, standard error of the mean. ^a,b^ Within a row means followed by different letters are different at *p* < 0.05. ^1^ There were 12 replicates per treatment.

**Table 4 animals-12-01271-t004:** Effects of two supplementary doses of vitamin A on length and relative organ weight of intestine in young laying pullets ^1^.

Item	Control	Treatment	SEM	*p*-Value
D 14				
Length, cm				
Duodenum	12.78 ^b^	13.81 ^a^	0.31	0.04
Jejunum	30.94	32.33	0.87	0.24
Ileum	27.24 ^b^	28.83 ^a^	0.80	0.04
Relative organ weight, %				
Duodenum	1.81	1.85	0.03	0.07
Jejunum	2.29	2.35	0.05	0.37
Ileum	1.79	1.82	0.03	0.65
D 21				
Length, cm				
Duodenum	15.38	15.72	0.35	0.42
Jejunum	32.23 ^b^	34.04 ^a^	0.90	0.04
Ileum	30.02	30.18	0.88	0.66
Relative organ weight, %				
Duodenum	1.69 ^b^	1.78 ^a^	0.03	0.02
Jejunum	2.06	2.11	0.06	0.68
Ileum	1.30	1.32	0.03	0.36
D 28				
Length, cm				
Duodenum	15.55 ^b^	16.50 ^a^	0.38	0.04
Jejunum	33.61 ^b^	35.58 ^a^	0.92	0.02
Ileum	29.65	30.14	0.85	0.55
Relative organ weight, %				
Duodenum	1.36	1.40	0.03	0.29
Jejunum	1.44 ^b^	1.62 ^a^	0.05	0.01
Ileum	1.05	1.07	0.04	0.67
D 35				
Length, cm				
Duodenum	16.67 ^b^	17.90 ^a^	0.40	<0.01
Jejunum	37.20 ^b^	40.16 ^a^	1.01	0.02
Ileum	33.47 ^b^	38.05 ^a^	0.99	<0.01
Relative organ weight, %				
Duodenum	1.34 ^b^	1.48 ^a^	0.03	<0.01
Jejunum	1.66 ^b^	1.87 ^a^	0.04	0.02
Ileum	1.46 ^b^	1.67 ^a^	0.03	0.01

SEM, standard error of the mean. ^a,b^ Within a row means followed by different letters are different at *p* < 0.05. ^1^ There were 12 replicates per treatment.

**Table 5 animals-12-01271-t005:** Effects of two supplementary doses of vitamin A on intestinal morphology of young laying pullets ^1^.

Item	Control	Treatment	SEM	*p*-Value
D 14				
Jejunum				
Villus height, μm	535	575	33	0.33
Crypt depth, μm	121	104	11	0.41
VCR	4.42 ^b^	5.53 ^a^	0.42	0.02
Ileum				
Villus height, μm	343 ^b^	396 ^a^	21	0.03
Crypt depth, μm	97	86	10	0.21
VCR	3.54 ^b^	4.60 ^a^	0.40	0.01
D 21				
Jejunum				
Villus height, μm	611	678	35	0.09
Crypt depth, μm	117	107	10	0.63
VCR	5.22 ^b^	6.34 ^a^	0.48	0.03
Ileum				
Villus height, μm	361 ^b^	412 ^a^	20	0.02
Crypt depth, μm	82	78	8	0.74
VCR	4.40 ^b^	5.28 ^a^	0.41	0.04
D 28				
Jejunum				
Villus height, μm	830	883	33	0.08
Crypt depth, μm	125	120	11	0.84
VCR	6.64	7.36	0.43	0.10
Ileum				
Villus height, μm	503 ^b^	562 ^a^	22	<0.01
Crypt depth, μm	88	86	8	0.85
VCR	5.72 ^b^	6.53 ^a^	0.38	0.03
D 35				
Jejunum				
Villus height, μm	901 ^b^	1032 ^a^	36	<0.01
Crypt depth, μm	150	148	10	0.63
VCR	6.01 ^b^	6.97 ^a^	0.42	0.01
Ileum				
Villus height, μm	515 ^b^	586 ^a^	25	<0.01
Crypt depth, μm	96	86	9	0.77
VCR	5.36 ^b^	6.81 ^a^	0.40	<0.01

SEM, standard error of the mean. VCR, villus height to crypt depth ratio. ^a,b^ Within a row means followed by different letters are different at *p* < 0.05. ^1^ There were 12 replicates per treatment.

**Table 6 animals-12-01271-t006:** Effects of two supplementary doses of vitamin A on relative organ weight (%) of immune organs in young laying pullets ^1^.

Item	Control	Treatment	SEM	*p*-Value
D 14				
Spleen	0.17 ^b^	0.21 ^a^	0.01	0.01
Thymus	0.61 ^b^	0.69 ^a^	0.03	0.01
Bursa	0.43	0.44	0.03	0.89
D 21				
Spleen	0.21 ^b^	0.26 ^a^	0.02	<0.01
Thymus	0.73 ^b^	0.79 ^a^	0.03	0.04
Bursa	0.53	0.56	0.03	0.44
D 28				
Spleen	0.25	0.26	0.02	0.63
Thymus	0.63 ^b^	0.70 ^a^	0.03	0.04
Bursa	0.61 ^b^	0.68 ^a^	0.03	0.03
D 35				
Spleen	0.27	0.27	0.01	0.51
Thymus	0.66	0.69	0.02	0.24
Bursa	0.61 ^b^	0.66 ^a^	0.02	0.02

SEM, standard error of the mean. ^a,b^ Within a row means followed by different letters are different at *p* < 0.05. ^1^ There were 12 replicates per treatment.

## Data Availability

The study did not report any data.

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
