# Peer review of "A Comparison of Two Supplementary Doses of Vitamin A on Performance, Intestine and Immune Organ Development, as well as Gene Expression of Inflammatory Factors in Young Hy-Line Brown Laying Pullets"

_animals, 2022, doi:10.3390/ani12101271_

Round 1

Reviewer 1 Report

Comment 1: The manuscript presents results from comparing two treatments; however, all statements assume they tested two treatments. It is impossible to infer "increasing levels " when they tested only two levels.

Comment 2: One of the levels of vitamin A is 40% below the genetic line nutritional recommendation, and the other is 20% higher. The current recommendation of the specific genetic line used is 10000 UI/kg as available in their management guides and is well known by the whole industry and academia. Therefore, inferring that the higher level provides benefits is a highly biased inference (and conclusion). In reality, the authors tested the effect of providing a vitamin A level way below the standard recommendation.

Comment 3: Currently, the industry does not use NRC1994 nutritional values; therefore, inferences testing those levels lack real value.

Comment 4: The information you have gathered is interesting, and a manuscript with it might work, but it would need to be redesigned the opposite way, as per the comments above.

Author Response

Comment 1: The manuscript presents results from comparing two treatments; however, all statements assume they tested two treatments. It is impossible to infer "increasing levels " when they tested only two levels.

Response: Thank you very much for your comment. We agree with your comment. We have revised the title and sentences in the manuscript to “A comparison of two supplementary doses of vitamin A on the performance, intestine and immune organs development as well as gene expressions of inflammatory factors in young Hy-Line Brown laying pullets”.

Comment 2: One of the levels of vitamin A is 40% below the genetic line nutritional recommendation, and the other is 20% higher. The current recommendation of the specific genetic line used is 10000 UI/kg as available in their management guides and is well known by the whole industry and academia. Therefore, inferring that the higher level provides benefits is a highly biased inference (and conclusion). In reality, the authors tested the effect of providing a vitamin A level way below the standard recommendation.

Response: Thank you very much for your comment. We agree with your comment. The current recommendation of VA dose is indeed 10000 IU/kg for young Hy-Line Brown hens. However, the Chinese Safe Use Standard of Feed Additives (No. 2625) stipulates that the maximum dose of VA for young pullets after 14 days is 10000 IU/kg. Before the current experiment, we collected several commercial pullets feed samples and determined the VA concentrations. We found that the majority of commercial feed samples’ VA concentrations were about 6000 IU/kg. So according to the above cases, we designed the two doses for 6000 and 12000 IU/kg, respectively. We agree with your comment and we will improve the experimental design in the next trial. We also have supplemented the explanation for the designed VA doses in the Discussion section. Thank you again for your comment.

Comment 3: Currently, the industry does not use NRC1994 nutritional values; therefore, inferences testing those levels lack real value.

Response: Thank you very much for your comment. The recommended dose for young laying pullets in NRC (1994) is 1420 IU/kg which is too low to use under practical conditions. So we according to the Chinese Safe Use Standard of Feed Additives (No. 2625) and commercial feed samples’ VA concentrations, we designed the two VA doses in the current experiment. We agree with your comment that we will explore a higher VA concentration effect in the future study.

Comment 4: The information you have gathered is interesting, and a manuscript with it might work, but it would need to be redesigned the opposite way, as per the comments above.

Response: Thank you very much for your comment. We have re-written the manuscript according to your above comments. The revised section was marked by red font in the revised version file. Thank you for your review work and hope for your reply.

Reviewer 2 Report

In the present paper I carefully reviewed, the Authors aimed to investigate the effects of increasing supplementary doses of vitamin A on performance, gut development e and immune organs as well as gene expressions of inflammatory factors in Hy-Line Brown laying pullets.

I would like to congratulate Authors for the good-quality of their article, the literature reported used to write the paper, and for the clear and appropriate structure.

The manuscript is well written, presented and discussed, and understandable to a specialist readership.

In general, the organization and the structure of the article are satisfactory and in agreement with the journal instructions for authors. The subject is adequate with the overall journal scope.

The work shows a conscientious study in which a very exhaustive discussion of the literature available has been carried out.

The Introduction section provides sufficient background, and the other sections include results clearly presented and analyzed exhaustively.

However, as specific comments, with the aim to further improve the quality of the paper.

The Introduction section could be further improved by adding a couple of sentences referring to recently published papers.

The Conclusion section could be improved; also, the Authors have to check if alle references have been cited in the text.

Check the style of the references list according to the journal's guidelines. 

Tables 5 and 6: add the units used for measurements.

So, I recommend the acceptance of the paper after revision.

Author Response

In the present paper I carefully reviewed, the Authors aimed to investigate the effects of increasing supplementary doses of vitamin A on performance, gut development and immune organs as well as gene expressions of inflammatory factors in Hy-Line Brown laying pullets.

I would like to congratulate Authors for the good-quality of their article, the literature reported used to write the paper, and for the clear and appropriate structure.

The manuscript is well written, presented and discussed, and understandable to a specialist readership.

In general, the organization and the structure of the article are satisfactory and in agreement with the journal instructions for authors. The subject is adequate with the overall journal scope.

The work shows a conscientious study in which a very exhaustive discussion of the literature available has been carried out.

The Introduction section provides sufficient background, and the other sections include results clearly presented and analyzed exhaustively.

However, as specific comments, with the aim to further improve the quality of the paper.

The Introduction section could be further improved by adding a couple of sentences referring to recently published papers.

Response: Thank you very much for your comment. We have added several sentences as well as new references according to your suggestion.

The Conclusion section could be improved; also, the Authors have to check if all references have been cited in the text.

Check the style of the references list according to the journal's guidelines.

Response: Thank you very much for your comment. We have revised the Conclusion section and checked the references and style according to your suggestion.

Tables 5 and 6: add the units used for measurements.

Response: Thank you very much for your comment. We have added the units in these tables according to your suggestion.

So, I recommend the acceptance of the paper after revision.

Response: Thank you very much for your comment. Hope for your reply.

Round 2

Reviewer 1 Report

Good job.